# An Entropy-Based Tool to Help the Interpretation of Common-Factor Spaces in Factor Analysis

**DOI:** 10.3390/e23020140

**Published:** 2021-01-24

**Authors:** Nobuoki Eshima, Claudio Giovanni Borroni, Minoru Tabata, Takeshi Kurosawa

**Affiliations:** 1Center for Educational Outreach and Admissions, Kyoto University, Yoshida-machi, Sakyoku, Kyoto 660-8501, Japan; 2Department of Statistics and Quantitative Methods, University of Milano Bicocca, 20126 Milano, Italy; claudio.borroni@unimib.it; 3Department of Mathematical Sciences, Osaka Prefecture University, Osaka 599-8532, Japan; mnrtabata@luck.ocn.ne.jp; 4Department of Applied Mathematics, Tokyo University of Science, Kagurazaka, Shinzyukuku, Tokyo 162-0825, Japan; tkuro@rs.tus.ac.jp

**Keywords:** canonical factor analysis, canonical common factor, entropy, factor contribution

## Abstract

This paper proposes a method for deriving interpretable common factors based on canonical correlation analysis applied to the vectors of common factors and manifest variables in the factor analysis model. First, an entropy-based method for measuring factor contributions is reviewed. Second, the entropy-based contribution measure of the common-factor vector is decomposed into those of canonical common factors, and it is also shown that the importance order of factors is that of their canonical correlation coefficients. Third, the method is applied to derive interpretable common factors. Numerical examples are provided to demonstrate the usefulness of the present approach.

## 1. Introduction

In factor analysis, extracting interpretable factors is important for practical data analysis. In order to carry it out, methods for factor rotation have been studied, e.g., varimax [1] and orthomax [2] for orthogonal rotations and oblimin [3] and orthoblique [4] for oblique rotations. The basic idea for factor rotation in factor analysis is owed to the criteria of simple structures of factor analysis models by Thurstone [5], and the methods of factor rotation are constructed with respect to maximizations of variations of the squared factor loadings in order to derive simple structures of factor analysis models. Let $X_{i}$ be manifest variables, let $\xi_{j}$ be latent variables (common factors), let $\varepsilon_{i}$ be unique factors related to $X_{i}$, and finally, let $\lambda_{ij}$ be factor loadings that are weights of common factors $\xi_{j}$ to explain $X_{i}$. Then, the factor analysis model is given as follows:(1)$$X_{i}=\sum_{j=1}^{m}\lambda_{ij}\xi_{j}+\varepsilon_{i},\;\;\;i=1,2,\ldots ,p,$$
where$$\{\begin{array}{c} E\left(X_{i}\right)=E\left(\varepsilon_{i}\right)=0,\;i=1,2,\ldots ,p;\\ E\left(\xi_{j}\right)=0,\;j=1,2,\ldots ,m;\\ \mathrm{Var}\left(\xi_{j}\right)=1,\;j=1,2,\ldots ,m;\;\mathrm{Cov}\left(\xi_{k},\xi_{l}\right)=\phi_{kl};\\ \mathrm{Var}\left(\varepsilon_{i}\right)=\omega_{i}^{2}>0,\;i=1,2,\ldots ,p;\\ \mathrm{Cov}\left(\varepsilon_{k},\varepsilon_{l}\right)=0,\;k\neq l. \end{array}$$
To derive simple structures of factor analysis models, for example, in the varimax method, the following variation function of squared factor loadings is maximized with respect to factor loadings:(2)$$V=\overset{p}{\underset{i=1}{\sum }}\overset{m}{\underset{j=1}{\sum }}{\left(\lambda_{ij}^{2}-\bar{\lambda^{2}}\right)}^{2},$$
where $\bar{\lambda^{2}}=\frac{1}{pm}\sum_{i=1}^{p}\sum_{j=1}^{m}\lambda_{ij}^{2}$. In this sense, the basic factor rotation methods can be viewed as those for exploratively analyzing multidimensional common-factor spaces. The interpretation of factors is made according to manifest variables with large weights in common factors. As far as we know, novel methods for factor rotation have not been investigated except for rotation methods similar to the above basic ones. In real data analyses, manifest variables are usually classified into some groups of variables in advance that may have common factors and concepts for themselves. For example, suppose we have a test battery including the following five subjects: Japanese, English, Social Science, Mathematics, and Natural Science. It is then reasonable to classify the five subjects into two groups, {Japanese, English, Social Science} and {Mathematics, Natural Science}. In such cases, it is meaningful to determine common factors related to the two manifest variable groups. For this objective, it is useful to develop a novel method to derive the common factors based on a factor contribution measure. In conventional methods of factor rotation, for example, as mentioned above, variation function (2) for the varimax method is not related to factor contribution.

An entropy-based method for measuring factor contribution was proposed by [6], and the method can measure factor contributions to manifest variables vectors and can decompose the factor contributions into those of manifest subvectors and individual manifest variables. By using the method, we can derive important common factors related to the manifest subvectors and the manifest variables. The aim of the present paper is to propose a new method for deriving simple structures based on entropy, that is, extracting common factors easy to interpret. In Section 2, an entropy-based method for measuring factor contribution [6] is reviewed to apply its properties for deriving simple structures in factor analysis models. Section 3 discusses canonical correlation analysis between common factors and manifest variables, and the contributions of common factors to the manifest variables are decomposed into components related to the extracted pairs of canonical variables. A numerical example is given to demonstrate the approach. In Section 4, canonical correlation analysis is applied to obtain common factors easy to interpret, and the contributions of the extracted factors are measured. Numerical examples are given to illustrate the present approach, and finally, Section 5 provides discussions and conclusions to summarize the present approach.

## 2. Entropy-Based Method for Measuring Factor Contributions

First, in order to derive factor contributions, factor analysis model (1) with error terms $\varepsilon_{i},\;i=1,2,\ldots ,p$, which are normally distributed, can be discussed in the framework of generalized linear models (GLMs) [7]. A general path diagram among manifest variables $X_{i},\;i=1,2,\ldots ,p$ and common factors $\xi_{j},\;j=1,2,\ldots ,m$ in the factor analysis model is illustrated in Figure 1. The conditional density functions of manifest variables of $X_{i},\;i=1,2,\ldots ,p$, given the factors $\xi_{j},\;j=1,2,\ldots ,m$, are expressed as follows:$$\begin{array}{c} f_{i}\left(x_{i}|\xi \right)=\frac{1}{\sqrt{2\pi \omega_{i}^{2}}}\exp\left(-\frac{{\left(x_{i}-\sum_{j=1}^{m}\lambda_{ij}\xi_{j}\right)}^{2}}{2\omega_{i}^{2}}\right)\\ =\exp\left(\frac{x_{i}\sum_{j=1}^{m}\lambda_{ij}\xi_{j}-\frac{1}{2}{\left(\sum_{j=1}^{m}\lambda_{ij}\xi_{j}\right)}^{2}}{\omega_{i}^{2}}-\frac{x_{i}^{2}}{2\omega_{i}^{2}}-\log\sqrt{2\pi \omega_{i}^{2}}\right),\;i=1,2,\ldots ,p. \end{array}$$
Let $\theta_{i}=\sum_{j=1}^{m}\lambda_{ij}\xi_{j}$ and $d\left(x_{i},\omega_{i}^{2}\right)=-\frac{x_{i}^{2}}{2\omega_{i}^{2}}-\log\sqrt{2\pi \omega_{i}^{2}}$. Then, the above density function is described in a GLM framework as
(3)$$f_{i}\left(x_{i}|\xi \right)=\exp\left(\frac{x_{i}\theta_{i}-\frac{1}{2}\theta_{i}^{2}}{\omega_{i}^{2}}+d\left(x_{i},\omega_{i}^{2}\right)\right),\;i=1,2,\ldots ,p.$$
According to the local independence of the manifest variables in factor analysis model (1), the conditional density function of $X={\left(X_{1},X_{2},\ldots ,X_{p}\right)}^{T}$ given $\xi ={\left(\xi_{1},\;\xi_{2},\ldots ,\xi_{m}\right)}^{T}$ is expressed as
(4)$$f\left(x|\xi \right)=\overset{p}{\underset{i=1}{\prod }}\exp\left(\frac{x_{i}\theta_{i}-\frac{1}{2}\theta_{i}^{2}}{\omega_{i}^{2}}+d\left(x_{i},\omega_{i}^{2}\right)\right)=\exp\left(\sum_{i=1}^{p}\frac{x_{i}\theta_{i}-\frac{1}{2}\theta_{i}^{2}}{\omega_{i}^{2}}+\sum_{i=1}^{p}d\left(x_{i},\omega_{i}^{2}\right)\right).$$

Let g(ξ) be the joint density function of common-factor vector $={\left(\xi_{1},\;\xi_{2},\ldots ,\xi_{m}\right)}^{T}$; let $f_{i}\left(x_{i}\right)$ be the marginal density functions of $X_{i},\;i=1,2,\ldots ,p$; and let us set
(5)$$\mathrm{KL}\left(X,\;\xi \right)=\iint^{\;}f\left(x|\xi \right)g\left(\xi \right)\log\frac{f\left(x|\xi \right)}{f\left(x\right)}dxd\xi +\iint^{\;}f\left(x\right)g\left(\xi \right)\log\frac{f\left(x\right)}{f\left(x|\xi \right)}dxd\xi ,$$
(6)$$\begin{array}{c} \mathrm{KL}\left(X_{i},\xi \right)=\iint^{\;}f_{i}\left(x_{i}|\xi \right)g\left(\xi \right)\log\frac{f_{i}\left(x_{i}|\xi \right)}{f_{i}\left(x_{i}\right)}dx_{i}d\xi +\iint^{\;}f_{i}\left(x_{i}\right)g\left(\xi \right)\log\frac{f_{i}\left(x_{i}\right)}{f_{i}\left(x_{i}|\xi \right)}dx_{i}d\xi ,\;\\ i=1,2,\ldots ,p. \end{array}$$
where “KL” stands for “Kullback–Leibler information” [8]. From (3) and (4), we have
(7)$$\mathrm{KL}\left(X_{i},\xi \right)=\frac{\mathrm{Cov}\left(X_{i},\theta_{i}\right)}{\omega_{i}^{2}}=\overset{m}{\underset{j=1}{\sum }}\frac{\lambda_{ij}\mathrm{Cov}\left(X_{i},\xi_{j}\right)}{\omega_{i}^{2}},i=1,2,\ldots ,p;$$
(8)$$\mathrm{KL}\left(X,\;\xi \right)=\overset{p}{\underset{i=1}{\sum }}\frac{\mathrm{Cov}\left(X_{i},\theta_{i}\right)}{\omega_{i}^{2}}=\overset{p}{\underset{i=1}{\sum }}\overset{m}{\underset{j=1}{\sum }}\frac{\lambda_{ij}\mathrm{Cov}\left(X_{i},\xi_{j}\right)}{\omega_{i}^{2}}.$$
The above quantities (7) and (8) are interpreted as the signal-to-noise ratios for dependent variables $X_{i}$ and predictors $\theta_{i}$; and the signal-to-noise ratio for dependent-variable vectors X and common-factor vector ξ, respectively. 

From (7) and (8), the following theorem can be derived [6]:

**Theorem** **1.**
*In factor analysis model (1), let *
$X={\left(X_{1},\;X_{2},\ldots ,X_{p}\right)}^{T}$
*and*
$\xi ={\left(\xi_{1},\;\xi_{2},\ldots ,\xi_{m}\right)}^{T}$
*. Then,*
$$\mathrm{KL}\left(X,\xi \right)=\overset{p}{\underset{i=1}{\sum }}\mathrm{KL}\left(X_{i},\xi \right).$$


Consistently, the following theorem, which is actually an extended version of Corollary 1 in [6], can be also obtained:

**Theorem** **2.**
*Let manifest variable subvectors *
$X^{\left(a\right)},\;a=1,2,\ldots ,A$
*be any decomposition of manifest variable vector*
$X={\left(X_{1},\;X_{2},\ldots ,X_{p}\right)}^{T}$
*. Then,*
(9)$$\mathrm{KL}\left(X,\xi \right)=\overset{A}{\underset{a=1}{\sum }}\mathrm{KL}\left(X^{\left(a\right)},\xi \right).$$


Following Eshima et al. [6], the contribution of factor vector $\xi ={\left(\xi_{1},\;\xi_{2},\ldots ,\xi_{m}\right)}^{T}$ to manifest variable vector $X={\left(X_{1},\;X_{2},\ldots ,X_{p}\right)}^{T}$ is thus defined as
C(ξ→X)=KL(X,ξ),
so that, in Theorem 2, the contributions of factor vector $\xi ={\left(\xi_{1},\;\xi_{2},\ldots ,\xi_{m}\right)}^{T}$ to manifest variable vectors $X^{\left(a\right)},\;a=1,2,\ldots ,A$ are defined by
$$C\left(\xi \to X^{\left(a\right)}\right)=\mathrm{KL}\left(X^{\left(a\right)},\xi \right),\;a=1,2,\ldots ,A.$$
Let $\xi^{\backslash j}$ be subvectors of all variables $\xi_{i}$ except $\xi_{j}$ from $\xi ={\left(\xi_{1},\;\xi_{2},\ldots ,\xi_{m}\right)}^{T}$, i.e.,
$$\xi^{\backslash j}={\left(\xi_{1},\;\xi_{2},\ldots ,\xi_{j-1},\xi_{j+1},\ldots ,\xi_{m}\right)}^{T},\;j=1,2,\ldots ,m;$$
and let $\mathrm{KL}\left(X,\xi^{\backslash j}|\xi_{j}\right)$ and $\mathrm{KL}\left(X^{\left(a\right)},\xi^{\backslash j}|\xi_{j}\right)$ be the conditional Kullback–Leibler information as defined in (5) and (6). The contributions of common factors $\xi_{j}$ are defined by
$$C\left(\xi_{j}\to X\right)=\mathrm{KL}\left(X,\xi \right)-\;\mathrm{KL}\left(X,\xi^{\backslash j}|\xi_{j}\right),$$
$$C\left(\xi_{j}\to X^{\left(a\right)}\right)=\mathrm{KL}\left(X^{\left(a\right)},\xi \right)-\mathrm{KL}\left(X^{\left(a\right)},\xi^{\backslash j}|\xi_{j}\right),\;j=1,2,\ldots ,m.$$

**Remark** **1.**
*Information*
$\mathrm{KL}\left(X,\xi^{\backslash j}|\xi_{j}\right)$
*and*
$\mathrm{KL}\left(X^{\left(a\right)},\xi^{\backslash j}|\xi_{j}\right)$
*can be expressed by using the conditional covariances*
$\mathrm{Cov}\left(X_{i},\theta_{i}|\xi_{j}\right)$
*. For example,*
$$\mathrm{KL}\left(X,\xi^{\backslash j}|\xi_{j}\right)=\overset{p}{\underset{i=1}{\sum }}\frac{\mathrm{Cov}\left(X_{i},\theta_{i}|\xi_{j}\right)}{\omega_{i}^{2}}.$$


Finally, the following decomposition of KL(X,ξ) holds for orthogonal factors ([6], Theorem 3):

**Theorem** **3.**
*If the common factors are mutually independent, it follows that*
$$C\left(\xi \to X\right)=\overset{m}{\underset{j=1}{\sum }}\overset{A}{\underset{a=1}{\sum }}C\left(\xi_{j}\to X^{\left(a\right)}\right)=\sum_{j=1}^{m}\sum_{i=1}^{p}C\left(\xi_{j}\to X_{i}\right).$$


The entropy coefficient of determination (ECD) [9] between ξ and X is defined by
$$\mathrm{ECD}\left(\xi ,X\right)=\frac{\mathrm{KL}\left(\xi ,X\right)}{\mathrm{KL}\left(\xi ,X\right)+1},$$
so that the total relative contribution of factor vector ξ to manifest variable vector X in entropy can be defined as
$$\tilde{\mathrm{RC}}\left(\xi \to X\right)=\mathrm{ECD}\left(\xi ,X\right)=\frac{C\left(\xi \to X\right)}{C\left(\xi \to X\right)+1},$$
while, for a single factor $\xi_{j}$, two relative contribution ratios can be defined:$$\begin{array}{c} \mathrm{RC}\left(\xi_{j}\to X\right)=\frac{C\left(\xi_{j}\to X\right)}{C\left(\xi \to X\right)}=\frac{\mathrm{KL}\left(X,\xi \right)-\;\mathrm{KL}\left(X,\xi^{\backslash j}|\xi_{j}\right)}{\mathrm{KL}\left(\xi ,X\right)},\\ \tilde{\mathrm{RC}}\left(\xi_{j}\to X\right)=\frac{C\left(\xi_{j}\to X\right)}{\mathrm{KL}\left(\xi ,X\right)+1}=\frac{\mathrm{KL}\left(X,\xi \right)-\;\mathrm{KL}\left(X,\xi^{\backslash j}|\xi_{j}\right)}{\mathrm{KL}\left(\xi ,X\right)+1} \end{array}$$
(see [6] for details).

Second, factor analysis model (1) in a general case is discussed. Let Σ be the variance–covariance matrix of manifest variable vector $X={\left(X_{1},X_{2},\ldots ,X_{p}\right)}^{T}$; let Ω be the p×p variance–covariance matrix of unique factor vector $\varepsilon ={\left(\varepsilon_{1},\varepsilon_{2},\ldots ,\varepsilon_{p}\right)}^{T}$; let Λ be the p×m factor loading matrix of $\lambda_{ij}$; and let Φ be the correlation matrix of common-factor vector $\xi ={\left(\xi_{1},\xi_{2},\ldots ,\xi_{m}\right)}^{T}$. Then, model (1) can be expressed as
X=Λξ+ε
and we have
$$\Sigma =\Lambda \Phi \Lambda^{T}+\Omega .$$
Now, the above discussion is extended in a general factor analysis model (1) with the following variance–covariance matrix of X and ε:(10)$$\left(\begin{array}{cc} \Lambda \Phi \Lambda^{T}+\Omega & \Lambda \Phi \\ \Phi \Lambda^{T} & \Phi \end{array}\right).$$
Let θ=Λξ be the predictor vector of manifest variable vector $X^{T}=\left(X_{1},X_{2},\ldots ,X_{p}\right)$. Then, the contribution of common-factor vector ξ to manifest variable vector X is defined by the following generalized signal-to-noise ratio:(11)$$E\left(X^{T}\Omega^{-1}\theta \right)=\frac{E\left(X^{T}\tilde{\Omega }\Lambda \xi \right)}{\left|\Omega \right|}=\frac{\mathrm{tr}\tilde{\Omega }\Lambda \Phi \Lambda^{T}}{\left|\Omega \right|},$$
where $\tilde{\Omega }$ is the cofactor matrix of Ω. The signal is $\mathrm{tr}\tilde{\Omega }\Lambda \Phi \Lambda^{T}$ and the noise |Ω|, and both are positive. Hence, the above quantity is defined as the explained entropy with the factor analysis model, and the same notation KL(X, ξ) as above is used, having to do with the Kullback–Leibler information for the factor analysis model with normal distribution errors (4). Similarly, in the general model, as in (9), signal-to-noise ratio (11) is decomposed into
$$\frac{\mathrm{tr}\tilde{\Omega }\Lambda \Phi \Lambda^{T}}{\left|\Omega \right|}=\overset{p}{\underset{i=1}{\sum }}\frac{\mathrm{Cov}\left(X_{i},\theta_{i}\right)}{\omega_{i}^{2}}=\overset{p}{\underset{i=1}{\sum }}\overset{m}{\underset{j=1}{\sum }}\frac{\lambda_{ij}\mathrm{Cov}\left(X_{i},\xi_{j}\right)}{\omega_{i}^{2}},$$
so the above theorems hold true as well. Thus, the results mentioned above are applicable to factor analysis models with error terms with non-normal distributions.

## 3. Canonical Factor Analysis

In order to derive interpretable factors from the common-factor space, we propose taking advantage of the results of canonical correlation analysis applied to manifest variables and common factors. This approach can be referred to as “canonical factor analysis” [10]. In the factor analysis model (1), the variance–covariance matrix of $X={\left(X_{1},\;X_{2},\ldots ,X_{p}\right)}^{T}$ and $\xi ={\left(\xi_{1},\;\xi_{2},\ldots ,\xi_{m}\right)}^{T}$ is given by (10). Then, we have the following theorem:

**Theorem** **4.**
*For canonical correlation coefficients *
$\rho_{k},\;k=1,2,\ldots ,m$
*between*
X 
*and*
ξ
*in factor analysis model (1) with (10), it follows that*
$$\mathrm{KL}\left(X,\xi \right)=\overset{m}{\underset{j=1}{\sum }}\frac{\rho_{j}^{2}}{1-\rho_{j}^{2}}.$$


**Proof.** Let $B^{\left(1\right)}$, $B^{\left(2\right)}$, and F be m×p, (p−m)×p, and m×m matrices, respectively; let $V^{\left(1\right)}={\left(V_{1},V_{2},\ldots ,V_{m}\right)}^{T}=B^{\left(1\right)}X$, $V^{\left(2\right)}=B^{\left(2\right)}X$, and $\eta ={\left(\eta_{1},\eta_{2},\ldots ,\eta_{m}\right)}^{T}=F\xi$. It is assumed that $\left(V_{j},\eta_{j}\right)$ are the pairs of canonical variables with correlation coefficients $\rho_{j},\;j=1,2,\ldots ,m$; that matrices $\left(\begin{array}{c} B^{\left(1\right)}\\ B^{\left(2\right)} \end{array}\right)$ and F are nonsingular; and that $V^{\left(1\right)}$ and $V^{\left(2\right)}$ are statistically independent. Since all pairs of canonical variables $\left(V_{j},\eta_{j}\right)$ and $V^{\left(2\right)}$ are mutually independent, we have
$$\mathrm{KL}\left(V^{\left(2\right)},\eta \right)=0,\;\mathrm{KL}\left(V^{\left(1\right)},\eta_{j}\right)=\mathrm{KL}\left(V_{j},\eta_{j}\right),\;j=1,2,\ldots ,m.$$
From Theorem 2, it follows that
$$\begin{array}{c} \mathrm{KL}\left(X,\xi \right)=\mathrm{KL}\left(V,F\xi \right)=\mathrm{KL}\left(\left(\begin{array}{c} V^{\left(1\right)}\\ V^{\left(2\right)} \end{array}\right),\eta \right)=\mathrm{KL}\left(V^{\left(1\right)},\eta \right)+\mathrm{KL}\left(V^{\left(2\right)},\eta \right)=\mathrm{KL}\left(V^{\left(1\right)},\eta \right)=\overset{m}{\underset{j=1}{\sum }}\mathrm{KL}\left(V^{\left(1\right)},\eta_{j}\right)\\ =\overset{m}{\underset{j=1}{\sum }}\mathrm{KL}\left(V_{j},\eta_{j}\right)=\overset{m}{\underset{j=1}{\sum }}\frac{\rho_{j}^{2}}{1-\rho_{j}^{2}}. \end{array}$$
This completes the theorem. □

In the proof of the above theorem, we have
(12)$$\mathrm{KL}\left(X,\eta_{j}\right)=\mathrm{KL}\left(V_{j},\eta_{j}\right)=\frac{\rho_{j}^{2}}{1-\rho_{j}^{2}},\;j=1,2,\ldots ,m.$$
It implies that
$$C\left(\eta_{j}\to X\right)=C\left(\eta_{j}\to V_{j}\right)=\frac{\rho_{j}^{2}}{1-\rho_{j}^{2}};$$
$$\tilde{\mathrm{RC}}\left(\eta_{j}\to X\right)=\frac{\mathrm{KL}\left(X,\eta_{j}\right)}{\mathrm{KL}\left(X,\xi \right)+1}=\frac{\mathrm{KL}\left(X,\eta_{j}\right)}{\mathrm{KL}\left(\eta ,V\right)+1}=\frac{\frac{\rho_{j}^{2}}{1-\rho_{j}^{2}}}{\sum_{a=1}^{m}\frac{\rho_{a}^{2}}{1-\rho_{a}^{2}}+1};$$
$$\mathrm{RC}\left(\eta_{j}\to X\right)=\frac{\mathrm{KL}\left(X,\eta_{j}\right)}{\mathrm{KL}\left(\xi ,X\right)}=\frac{\mathrm{KL}\left(V_{j},\eta_{j}\right)}{\mathrm{KL}\left(V,\eta \right)}=\frac{\frac{\rho_{j}^{2}}{1-\rho_{j}^{2}}}{\sum_{a=1}^{m}\frac{\rho_{a}^{2}}{1-\rho_{a}^{2}}},\;j=1,2,\ldots ,m.\;$$
Theorem 4 shows that the contribution of common-factor vector ξ to manifest variable vector X is decomposed into those of canonical common factors $\eta_{j}$, i.e.,
$$\mathrm{KL}\left(X,\xi \right)=\overset{m}{\underset{j=1}{\sum }}\mathrm{KL}\left(X,\eta_{j}\right)=\overset{m}{\underset{j=1}{\sum }}\mathrm{KL}\left(V_{j},\eta_{j}\right),\;j=1,2,\ldots ,m.$$
Let us assume
(13)$$1>\rho_{1}^{2}\geq \rho_{2}^{2}\geq \ldots \geq \rho_{m}^{2}\geq 0.$$
According to the entropy-based criterion in Theorem 4, the order of importance of canonical common factors is that of canonical correlation coefficients. The interpretation of factors $\eta_{j}$ can be made with the corresponding manifest canonical variables $V_{j}$ and the factor loading matrix of canonical common factors η=Fξ. For the canonical common factors, the factor loading matrix can be obtained as $\Lambda^{*}=\Lambda F^{-1}$. We refer to the canonical correlation analysis in Theorem 4 as canonical factor analysis [10].

**Theorem** **5.**
*In factor analysis model (1), for any *
p×p
*and*
m×m
*nonsingular matrices*
P
*and*
Q
*, the canonical factor analysis between manifest variable vector*
PX
*and common-factor vector *
Qξ
*is invariant.*


**Proof.** Since the variance–covariance matrix of PX and Qξ is given by
$$\left(\begin{array}{cc} P & 0\\ 0 & Q \end{array}\right)\left(\begin{array}{cc} \Sigma & \Lambda^{T}\\ \Lambda & I_{m} \end{array}\right){\left(\begin{array}{cc} P & 0\\ 0 & Q \end{array}\right)}^{T},$$
the theorem follows. □

Notice that we also have
KL(PX,Qξ)=KL(X,ξ).
From the above theorem, the results of the canonical factor analysis do not depend on the initial common factors $\xi_{j}$ in factor analysis model (1). For factor analysis model (1), it follows that
$$\mathrm{KL}\left(X,\xi \right)=\overset{m}{\underset{j=1}{\sum }}\mathrm{KL}\left(V_{j},\eta_{j}\right)=\overset{p}{\underset{i=1}{\sum }}\mathrm{KL}\left(X_{i},\xi \right),$$
implying that
$$\frac{\mathrm{tr}\tilde{\Omega }\Lambda \Phi \Lambda^{T}}{\left|\Omega \right|}=\overset{m}{\underset{j=1}{\sum }}\frac{\rho_{j}^{2}}{1-\rho_{j}^{2}}\;=\overset{p}{\underset{i=1}{\sum }}\frac{R_{i}^{2}}{1-R_{i}^{2}},$$
where $R_{i}$ are the multiple correlation coefficients between manifest variables $X_{i}$ and factor vector $\xi =\left(\xi_{1},\xi_{2},\ldots ,\xi_{m}\right),\;i=1,2,\ldots ,p$. 

### Numerical Example 1

Table 1 shows the results of orthogonal factor analysis (varimax method by S-PLUS ver. 8.2) as reported in [6]; the same example is used here to demonstrate the canonical factor analysis mentioned above. In Table 1, manifest variables $X_{1},\;X_{2},\;\mathrm{and}\;X_{3}$ are scores in some subjects in the liberal arts, while variables $X_{4}\;\mathrm{and}\;X_{5}$ are those in the sciences. We refer to the factors as the initial common factors. In this example, from Table 1, the variance–covariance matrices in (10) are given as follows:
$$\Sigma =\left(\begin{array}{c} \begin{array}{ccccc} 1 & 0.54 & 0.39 & 0.42 & 0.36\\ 0.54 & 1 & 0.49 & 0.38 & 0.22\\ 0.39 & 0.49 & 1 & 0.21 & 0\\ 0.42 & 0.38 & 0.21 & 1 & 0.54\\ 0.36 & 0.22 & 0 & 0.54 & 1 \end{array} \end{array}\right),$$
$$\Phi =\left(\begin{array}{cc} 1 & 0\\ 0 & 1 \end{array}\right).$$
where covariance matrix $\Lambda^{T}$ is given in Table 1.

From the above matrices, to obtain the pairs of canonical variables, linear transformation matrices $B^{\left(1\right)}$ and F in Theorem 4 are as follows:$$B^{\left(1\right)}=\left(\begin{array}{cc} \begin{array}{cc} 0.19 & 0.20\\ 0.32 & 0.58 \end{array} & \begin{array}{ccc} 0.06 & 0.20 & 0.94\\ 0.37 & 0.00 & -0.65 \end{array} \end{array}\right),$$
and
$$F=\left(\begin{array}{cc} 0.32 & 0.95\\ 0.95 & -0.32 \end{array}\right).$$
By the above matrices, we have the following pairs of canonical variables $\left(V_{i},\eta_{i}\right)$ and their squared canonical correlation coefficients $\rho_{i}^{2}$:$$\{\begin{array}{c} V_{1}=0.19X_{1}+0.20X_{2}+0.06X_{3}+0.20X_{4}+0.94X_{5},\\ \eta_{1}=0.32\xi_{1}+0.95\xi_{2},\\ \rho_{1}^{2}=0.88, \end{array}$$
$$\{\begin{array}{c} V_{2}=0.32X_{1}+0.58X_{2}+0.37X_{3}+0.07X_{4}-0.65X_{5},\\ \eta_{2}=0.95\xi_{1}-0.32\xi_{2},\\ \rho_{2}^{2}=0.73. \end{array}$$
According to the above canonical variables, the factor loading for canonical factors $\eta_{i},\;i=1,2$ is calculated with the initial loading matrix Λ and the rotation matrix F, and we have
$$\begin{array}{c} \Lambda^{*T}={\left(\Lambda F^{-1}\right)}^{T}={\left(\begin{array}{cc} 0.32 & 0.95\\ 0.95 & -0.32 \end{array}\right)}^{-1}\left(\begin{array}{cc} \begin{array}{cc} 0.6 & 0.75\\ 0.39 & 0.24 \end{array} & \begin{array}{ccc} 0.65 & 0.32 & 0.00\\ 0.00 & 0.59 & 0.92 \end{array} \end{array}\right)\\ =\left(\begin{array}{cc} \begin{array}{cc} 0.56 & 0.47\\ 0.45 & 0.64 \end{array} & \begin{array}{ccc} 0.21 & 0.66 & 0.87\\ 0.62 & 0.12 & -0.29 \end{array} \end{array}\right). \end{array}$$
From the above results, the first canonical factor $\eta_{1}$ can be viewed as a general common ability (factor) to solve all five subjects. The second factor $\eta_{2}$ can be regarded as a factor related to subjects in the liberal arts, which is independent of the first canonical factor. In the canonical correlation analysis, the contributions of canonical factors are calculated. Since the multiple correlation coefficient between $\eta_{1}$ and $X={\left(X_{1},\;X_{2},\ldots ,X_{5}\right)}^{T}$ is $\rho_{1}^{2}=0.88$ and that between $\eta_{2}$ and X is $\rho_{2}^{2}=0.73$, we have
$$C\left(\eta_{1}\to X\right)=\frac{\rho_{1}^{2}}{1-\rho_{1}^{2}}=7.06,\;C\left(\eta_{2}\to X\right)=\frac{\rho_{2}^{2}}{1-\rho_{2}^{2}}=2.70.$$
Let $\xi =\left(\xi_{1},\xi_{2}\right)$. From the above results, we have
$$\begin{array}{c} C\left(\xi \to X\right)=\mathrm{KL}\left(\xi ,X\right)=C\left(\eta_{1}\to X\right)+\;C\left(\eta_{2}\to X\right)=9.86,\\ \tilde{\mathrm{CR}}\left(\xi \to X\right)=\frac{\mathrm{KL}\left(\xi ,X\right)}{\mathrm{KL}\left(\xi ,X\right)+1}=0.91\left(=\mathrm{ECD}\left(\xi ,X\right)\right). \end{array}$$
From this, 91% of the variation of manifest random vector X in entropy is explained by the common latent factors ξ. The contribution ratios of canonical common factors are calculated as follows:
$$\mathrm{CR}\left(\eta_{1}\to X\right)=\frac{7.06}{7.06+2.70}=0.72,\;\mathrm{CR}\left(\eta_{2}\to X\right)=2.70.$$
The contribution of the first canonical factor is about 2.6 times greater than that of the second one.

## 4. Deriving Important Common Factors Based on Decomposition of Manifest Variables into Subsets

From (9) in Theorem 2, KL(X,ξ) is decomposed into those for manifest variable subvectors $X^{\left(a\right)}$, $\mathrm{KL}\left(X^{\left(a\right)},\xi \right),\;a=1,2,\ldots ,A$. Thus, we have the following theorem:
**Theorem** **6.***Let manifest variable vector *X*be decomposed into subvectors*$X^{\left(a\right)},\;a=1,2,\ldots ,A$*. Let *$\rho_{\left(a\right)j},\;j=1,2,\ldots ,m_{\left(a\right)}$*be the canonical correlation coefficients between manifest variable subvector*$X^{\left(a\right)}$*and common-factor vector *ξ*,*a=1,2,…,A*in the factor analysis model (1), where *$m_{\left(a\right)}\leq \min\left\{\mathrm{dimension}\;\mathrm{of}\;X^{\left(a\right)},m\right\}$*. Then,*KL(X,ξ)*is decomposed into canonical components as follows:*$$\mathrm{KL}\left(X,\xi \right)=\overset{A}{\underset{a=1}{\sum }}\overset{m_{\left(a\right)}}{\underset{j=1}{\sum }}\frac{\rho_{\left(a\right)j}^{2}}{1-\rho_{\left(a\right)j}^{2}}.$$
**Proof.** For manifest variable vector $X^{\left(a\right)}$ and common-factor vector ξ, applying canonical correlation analysis, we have $m_{\left(a\right)}$ pairs of canonical variables $\left(V_{j}^{\left(\alpha \right)},\eta_{j}^{\left(\alpha \right)}\right)$ with squared canonical correlation coefficients $\rho_{\left(a\right)j}^{2},\;j=1,2,\ldots ,m_{\left(a\right)}$. Then, applying Theorem 4 to $\mathrm{KL}\left(X^{\left(a\right)},\xi \right)$ it follows that
$$\mathrm{KL}\left(X^{\left(a\right)},\xi \right)=\overset{m_{\left(a\right)}}{\underset{j=1}{\sum }}\mathrm{KL}\left(V_{j}^{\left(\alpha \right)},\eta_{j}^{\left(\alpha \right)}\right)=\overset{m_{\left(a\right)}}{\underset{j=1}{\sum }}\frac{\rho_{\left(a\right)j}^{2}}{1-\rho_{\left(a\right)j}^{2}},\;a=1,2,\ldots ,A.$$
From Theorem 2, the theorem follows. □

**Remark** **2.**
*As shown in the above theorem, the following relations hold:*
$$\mathrm{KL}\left(X^{\left(a\right)},\eta_{j}^{\left(\alpha \right)}\right)=\mathrm{KL}\left(V_{j}^{\left(\alpha \right)},\eta_{j}^{\left(\alpha \right)}\right)=\frac{\rho_{\left(a\right)j}^{2}}{1-\rho_{\left(a\right)j}^{2}},\;j=1,2,\ldots ,m_{\left(a\right)};\;a=1,2,\ldots ,A.$$


In this sense,
$$C\left(\eta_{j}^{\left(a\right)}\to X^{\left(a\right)}\right)=\frac{\rho_{\left(a\right)j}^{2}}{1-\rho_{\left(a\right)j}^{2}},\;j=1,2,\ldots ,m_{\left(a\right)};\;a=1,2,\ldots ,A.$$

To derive important common factors, the above theorem can be used. In many of the data in factor analysis, manifest variables can be classified into subsets that have common concepts (factors) to be measured. For example, in the data used for Table 1, it is meaningful to classify the five variables into two subsets $X^{\left(1\right)}=\left(X_{1},X_{2},X_{3}\right)$ and $X^{\left(2\right)}=\left(X_{4},X_{5}\right)$, where the first subset is related to the liberal arts and the second one is related to the sciences. In $\left(X^{\left(1\right)},\xi \right)$ and $\left(X^{\left(2\right)},\xi \right)$, it is possible to derive the latent ability for the liberal arts and that for the sciences, respectively.

### 4.1. Numerical Example 1 (Continued)

For $\left(X^{\left(1\right)},\xi \right)$ and $\left(X^{\left(2\right)},\xi \right)$, two sets of canonical variables are obtained, respectively, as follows:$$\{\begin{array}{c} \eta_{1}^{\left(1\right)}=0.95\xi_{1}+0.32\xi_{2},\;V_{1}^{\left(1\right)}=0.52X_{1}+0.76X_{2}+0.39X_{3},\;\rho_{\left(1\right)1}^{2}=0.77,\\ \eta_{2}^{\left(1\right)}=0.32\xi_{1}-0.95\xi_{2},\;V_{2}^{\left(1\right)}=0.71X_{1}-0.07X_{2}-0.71X_{3},\;\rho_{\left(1\right)2}^{2}=0.12, \end{array}$$
$$\{\begin{array}{c} \eta_{1}^{\left(2\right)}=0.06\xi_{1}+1.00\xi_{2},\;V_{1}^{\left(2\right)}=0.18X_{4}+0.98X_{5},\;\rho_{\left(2\right)1}^{2}=0.97,\\ \eta_{2}^{\left(2\right)}=1.00\xi_{1}-0.06\xi_{2},\;V_{2}^{\left(2\right)}=0.83X_{4}-0.55X_{5},\;\rho_{\left(2\right)2}^{2}=0.03. \end{array}$$
According to the above canonical variables, we have the following factor contributions:$$\{\begin{array}{c} C\left(\eta_{1}^{\left(1\right)}\to X^{\left(1\right)}\right)=C\left(\eta_{1}^{\left(1\right)}\to V_{1}^{\left(1\right)}\right)=\frac{0.77}{1-0.77}=3.27,\;\mathrm{CR}\left(\eta_{1}^{\left(1\right)}\to X^{\left(1\right)}\right)=0.96,\\ C\left(\eta_{2}^{\left(1\right)}\to X^{\left(1\right)}\right)=\frac{0.12}{1-0.12}=0.14.\;\mathrm{CR}\left(\eta_{2}^{\left(1\right)}\to X^{\left(1\right)}\right)=0.04; \end{array}$$
$$\{\begin{array}{c} C\left(\eta_{1}^{\left(2\right)}\to X^{\left(2\right)}\right)=6.14,\;\mathrm{CR}\left(\eta_{1}^{\left(2\right)}\to X^{\left(2\right)}\right)=0.97,\\ C\left(\eta_{2}^{\left(2\right)}\to X^{\left(2\right)}\right)=0.17,\;\mathrm{CR}\left(\eta_{2}^{\left(2\right)}\to X^{\left(2\right)}\right)=0.03. \end{array}$$
From the above results, canonical factors $\eta_{1}^{\left(1\right)}$ and $\eta_{1}^{\left(2\right)}$ can be interpreted as general common factors for the liberal arts and for the sciences, respectively. By using the factors, the factor loadings are given in Table 2. In this case, Table 2 is similar to Table 1; however, the factor analysis model is oblique and the correlation coefficient between $\eta_{1}^{\left(1\right)}$ and $\eta_{1}^{\left(2\right)}$ is 0.374. The contributions of the factors to manifest variable vector $X=\left(X_{1},X_{2},X_{3},X_{4},X_{5}\right)=\left(X^{\left(1\right)},X^{\left(2\right)}\right)$ are calculated as follows:$$\{\begin{array}{c} C\left(\eta_{1}^{\left(1\right)}\to X\right)=6.563,\;\mathrm{CR}\left(\eta_{1}^{\left(1\right)}\to X\right)=0.687,\tilde{\mathrm{CR}}\left(\eta_{1}^{\left(1\right)}\to X\right)=0.60,\;\\ C\left(\eta_{1}^{\left(2\right)}\to X\right)=4.223.\;\mathrm{CR}\left(\eta_{1}^{\left(2\right)}\to X\right)=0.442,\tilde{\mathrm{CR}}\left(\eta_{1}^{\left(2\right)}\to X\right)=0.39. \end{array}$$
In this case, factors $\eta_{1}^{\left(1\right)}$ and $\eta_{1}^{\left(2\right)}$ are correlated, so it follows that
$$\mathrm{CR}\left(\eta_{1}^{\left(1\right)}\to X\right)+\mathrm{CR}\left(\eta_{1}^{\left(2\right)}\to X\right)=1.129>1.$$

### 4.2. Numerical Example 2

Table 3 shows the results of the maximum likelihood factor analysis (orthogonal) for six scores $X_{i},\;i=1,2,\ldots ,6$ ([11], pp. 61–65); such results are treated as the initial estimates in the present analysis. In this example, variables are classified into the following three groups: variable $X_{1}$ is related to the Spearman’s g factor; variables $X_{2}$, $X_{3}$, and $X_{4}$ account for problem-solving ability; and variables $X_{5}$ and $X_{6}$ are associated with verbal ability [11]; however, it is difficult to explain the three factors by using Table 3. In this example, the present approach is employed for deriving the three factors. From (10) and Table 3, the correlation matrix of the manifest variables is given as follows:$$\hat{\Sigma }=\left(\begin{array}{cc} \begin{array}{ccc} 1 & 0.417 & 0.576\\ 0.417 & 1 & 0.567\\ 0.576 & 0.567 & 1 \end{array} & \begin{array}{ccc} 0.312 & 0.576 & 0.514\\ 0.306 & 0.265 & 0.263\\ 0.427 & 0.355 & 0.354 \end{array}\\ \begin{array}{ccc} 0.312 & 0.306 & 0.427\\ 0.576 & 0.265 & 0.355\\ 0.514 & 0.263 & 0.354 \end{array} & \begin{array}{ccc} 1 & 0.193 & 0.193\\ 0.193 & 1 & 0.799\\ 0.193 & 0.799 & 1 \end{array} \end{array}\right).$$
Let $X^{\left(2\right)}=\left(X_{2},X_{3},X_{4}\right)$, let $X^{\left(3\right)}=\left(X_{5},X_{6}\right)$, and let $\xi =\left(\xi_{1},\xi_{2}\right)$. Canonical correlation analysis is carried out for $\left(X_{1},\xi \right)$, $\left(X^{\left(2\right)},\xi \right)$, and $\left(X^{\left(3\right)},\xi \right)$, and we have the following canonical variables, respectively:$$\eta_{1}^{\left(1\right)}=\frac{1}{\sqrt{0.67^{2}+0.37^{2}}}\left(0.64\xi_{1}+0.37\xi_{2}\right)=0.87\xi_{1}+0.50\xi_{2},\;V_{1}^{\left(1\right)}=X_{1},\;\rho_{\left(1\right)1}^{2}=0.55$$
$$\{\begin{array}{c} \eta_{1}^{\left(2\right)}=0.52\xi_{1}+0.85\xi_{2},\;V_{1}^{\left(2\right)}=0.24X_{2}+0.96X_{3}+0.14X_{4},\;\rho_{\left(2\right)1}^{2}=0.83,\\ \eta_{2}^{\left(2\right)}=0.85\xi_{1}-0.52\xi_{2},\;V_{2}^{\left(2\right)}=0.81X_{2}-0.59X_{3}-0.02X_{4},\;\rho_{\left(1\right)2}^{2}=0.00, \end{array}$$
$$\{\begin{array}{c} \eta_{1}^{\left(3\right)}=0.99\xi_{1}-0.12\xi_{2},\;V_{1}^{\left(3\right)}=0.99X_{5}+0.11X_{6},\;\rho_{\left(3\right)1}^{2}=0.96,\\ \eta_{2}^{\left(3\right)}=0.12\xi_{1}+0.99\xi_{2},\;V_{2}^{\left(3\right)}=0.64X_{5}-0.77X_{6},\;\rho_{\left(3\right)2}^{2}=0.01. \end{array}$$
The contributions of canonical factors $\eta_{i}^{\left(k\right)},\;i=1,2;k=2.3$ are calculated as follows:$$\{\begin{array}{c} C\left(\eta_{1}^{\left(2\right)}\to X^{\left(2\right)}\right)=C\left(\eta_{1}^{\left(2\right)}\to V_{1}^{\left(2\right)}\right)=\frac{0.83}{1-0.83}=4.88,\;\mathrm{CR}\left(\eta_{1}^{\left(2\right)}\to X^{\left(2\right)}\right)=1.00,\\ C\left(\eta_{2}^{\left(2\right)}\to X^{\left(2\right)}\right)=\frac{0.00}{1-0.00}=0.00.\;\mathrm{CR}\left(\eta_{2}^{\left(1\right)}\to X^{\left(1\right)}\right)=0.00; \end{array}$$
$$\{\begin{array}{c} C\left(\eta_{1}^{\left(3\right)}\to X^{\left(3\right)}\right)=\frac{0.96}{1-0.96}=24.00,\;\mathrm{CR}\left(\eta_{1}^{\left(3\right)}\to X^{\left(3\right)}\right)=0.99,\\ C\left(\eta_{2}^{\left(2\right)}\to X^{\left(2\right)}\right)=0.01,\;\mathrm{CR}\left(\eta_{2}^{\left(2\right)}\to X^{\left(2\right)}\right)=0.01. \end{array}$$
The common factor $\eta_{1}^{\left(1\right)}\left(=g\right)$ can be interpreted as the Spearman’s g factor (general intelligence) and canonical common factors $\eta_{1}^{\left(2\right)}$ and $\eta_{1}^{\left(3\right)}$ can be interpreted as problem-solving ability and verbal ability, respectively. The correlation coefficients between the three factors are given by
$$\mathrm{Corr}\left(g,\eta_{1}^{\left(2\right)}\right)=0.88,\;\mathrm{Corr}\left(g,\eta_{1}^{\left(3\right)}\right)=0.80,\;\mathrm{Corr}\left(\eta_{1}^{\left(2\right)},\eta_{1}^{\left(3\right)}\right)=0.42.$$
The contributions of the above three factors to manifest variable vector $X=\left(X_{1},X_{2},X_{3},X_{4},X_{5},X_{6}\right)$ are computed as follows:$$\{\begin{array}{ccc} C\left(g\to X\right)=19.93, & \mathrm{CR}\left(g\to X\right)=0.68 & \tilde{\mathrm{CR}}\left(g\to X\right)=0.66\\ C\left(\eta_{1}^{\left(2\right)}\to X\right)=9.77, & \mathrm{CR}\left(\eta_{1}^{\left(2\right)}\to X\right)=0.33, & \tilde{\mathrm{CR}}\left(\eta_{1}^{\left(2\right)}\to X\right)=0.32,\\ C\left(\eta_{1}^{\left(3\right)}\to X\right)=25.02, & \mathrm{CR}\left(\eta_{1}^{\left(3\right)}\to X\right)=0.85, & \tilde{\mathrm{CR}}\left(\eta_{1}^{\left(3\right)}\to X\right)=0.82. \end{array}$$
The common-factor space is two-dimensional, and the factor loadings with common factors $\eta_{1}^{\left(2\right)}$ and $\eta_{1}^{\left(3\right)}$ are calculated as in Table 4. The table shows a clear interpretation of the common factors. Thus, the present method is effective for deriving interpretable factors in situations such as that of this example. The expressions of the factor analysis model can also be given by factor vectors $\left(g,\eta_{1}^{\left(2\right)}\right)$ and $\left(g,\eta_{1}^{\left(3\right)}\right)$, respectively. The present method is applicable for any subsets of manifest variables.

## 5. Discussion

In order to find interpretable common factors in factor analysis models, methods of factor rotation are often used. The methods are based on maximizations of variation functions of squares of factor loadings, and orthogonal or oblique factors are applied. The factors derived by the conventional methods may be interpretable; however, it may be more useful to propose a method for detecting interpretable common factors based on factor contribution measurement, i.e., importance of common factors. An entropy-based method for measuring factor contribution [6] can measure the contribution of the common-factor vector to the manifest variable vector, and one can decompose such a contribution into those of single manifest variables (Theorem 1) and into that of some manifest variable subvectors as well (Theorem 2). A characterization in the case of orthogonal factors can be also given (Theorem 3). The paper shows that the most important common factor with respect to entropy can be identified by using canonical correlation analysis between the factor vector and the manifest variable vector (Theorem 4). Theorem 4 shows that the contribution of the common-factor vector to the manifest variable vector can be decomposed into those of canonical factors and that the order of canonical correlation coefficients is that of factor contributions. In most multivariate data, manifest variables can be naturally classified into subsets according to common concepts as in Examples 1 and 2. By using Theorems 2 and 5, canonical correlation analysis can also be applied to derive canonical common factors from subsets of manifest variables and the initial common-factor vector (Theorem 6). According to the analysis, interpretable common factors can be obtained easily, as demonstrated in Examples 1 and 2. In Example 1, Table 1 and Table 2 have similar factor patterns; however, the derived factors in Table 1 are orthogonal and those in Table 2 are oblique. In Example 2, it may be difficult to interpret the factors in Table 3 produced by the varimax method. On the other hand, Table 4, obtained by using the present method, can be interpreted clearly. Finally, according to Theorem 5, the present method produces results that are invariant with respect to linear transformations of common factors, so that the method is independent of the initial common factors. The present method is the first one to derive interpretable factors based on a factor contribution measure, and the interpretable factors can be obtained easily through canonical correlation analysis between manifest variable subvectors and the factor vectors.

## Figures and Tables

**Figure 1 entropy-23-00140-f001:**
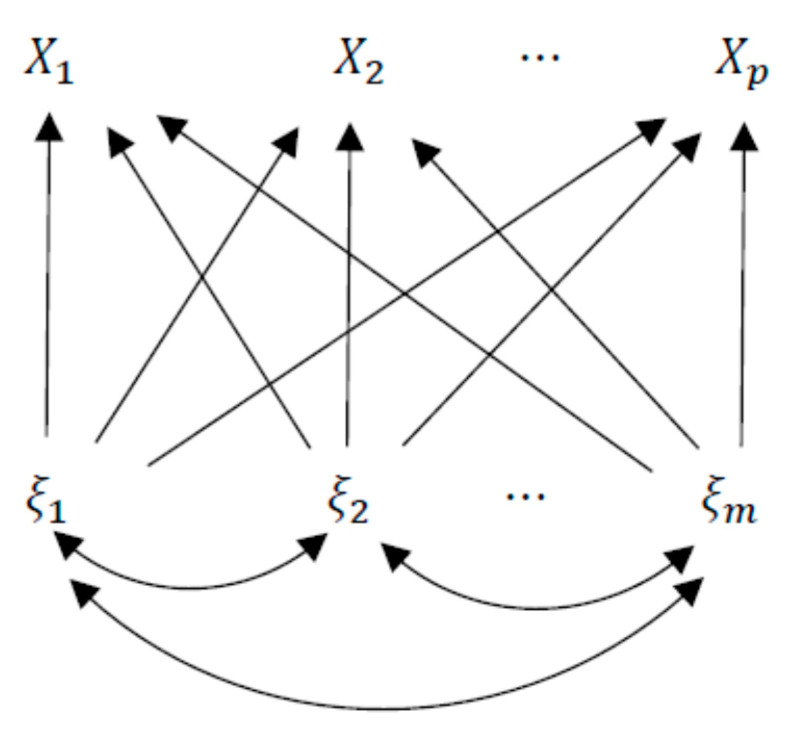
Path diagram of a general factor analysis model.

**Table 1 entropy-23-00140-t001:** Factor loadings of orthogonal (varimax) factor analysis.

	$X_{1}$	$X_{2}$	$X_{3}$	$X_{4}$	$X_{5}$
$\xi_{1}$	0.60	0.75	0.65	0.32	0.00
$\xi_{2}$	0.39	0.24	0.00	0.59	0.92
uniqueness	0.50	0.38	0.58	0.55	0.16

Uniqueness is the proportion of unique factor $\varepsilon_{i}$ related to manifest variable $X_{i}$.

**Table 2 entropy-23-00140-t002:** Factor loadings by using canonical common factors $\eta_{1}^{\left(1\right)}$ and $\eta_{1}^{\left(2\right)}$.

	$X_{1}$	$X_{2}$	$X_{3}$	$X_{4}$	$X_{5}$
$\eta_{1}^{\left(1\right)}$	0.62	0.80	0.70	0.31	−0.06
$\eta_{1}^{\left(2\right)}$	0.19	−0.02	−0.22	0.49	0.94
uniqueness	0.50	0.38	0.58	0.55	0.16

**Table 3 entropy-23-00140-t003:** The initial maximum likelihood estimates of factor loadings (varimax).

	$X_{1}$	$X_{2}$	$X_{3}$	$X_{4}$	$X_{5}$	$X_{6}$
$\xi_{1}$	0.64	0.34	0.46	0.25	0.97	0.82
$\xi_{2}$	0.37	0.54	0.76	0.41	−0.12	−0.03
uniqueness	0.45	0.59	0.21	0.77	0.04	0.33

**Table 4 entropy-23-00140-t004:** The factor loadings with common factors $\eta_{1}^{\left(2\right)}$ and $\eta_{1}^{\left(3\right)}$.

	$X_{1}$	$X_{2}$	$X_{3}$	$X_{4}$	$X_{5}$	$X_{6}$
$\eta_{1}^{\left(2\right)}$	0.49	0.63	0.89	0.48	−0.01	0.07
$\eta_{1}^{\left(3\right)}$	0.39	0.01	0.00	0.00	0.98	0.79
uniqueness	0.45	0.59	0.21	0.77	0.04	0.33

## Data Availability

Data sharing is not applicable to this article.

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
