# Peer review of "An Entropy-Based Tool to Help the Interpretation of Common-Factor Spaces in Factor Analysis"

_entropy, 2021, doi:10.3390/e23020140_

Round 1

Reviewer 1 Report

The topic is of great interest as the method can be used in a wide range of applications. However, I found a number of weaknesses that should be address for the manuscript to be published.

Main issues:

  • Introduction: It directly starts with equations, looks like a text book rather than a scientific article. The problem should be posed in the introduction, describing how this work deals with it. 
  • Review of literature/state of the art. It is simply missing. There are just 8 references cited, one of them from the own authors (by the way, replicating its theorems in this new work). The rest of the references are quite old, as a basic and stablished background. A review of the literature must be done, looking for recent works that have addressed the problem. After that, authors must make clear why this new tool contributes the state of the art.
  • Related to the references, in line 49: "the present theme". What is the present theme? Give details of the differences with the cited article?
  • Line 201 and elsewhere: The model is used when manifest variables can be classified into subsets. If this can be done a-priori, what is the real contribution of this method? A sort of confirmatory analysis? Moreover: How could this method be used for pattern discovery? I mean, when such variable classification cannot be done or it is not clear. That would be a nice contribution.
  •  

Regarding the model assumptions:

  • Normality assumption. In line 59, it is assumed that factors are normally distributed. Then, at the end of section 2, it is stated that results are applicable to non normal factors. I think this needs more justification, or cite a source for it.

Discussion:

  • Is the method implemented in any software? Do you plan to do it? (for example as an R package/function or alike?)
  • The two examples presented in the paper are quite similar. In this section further use cases might be mentioned, specially describing how the method would scale to many variables.

Further issues:

  • Line 30: Symbol \Sigma not defined
  • Figure 1: I think subindex X_m should be X_p
  • Line 67: KL needs citation.
  • Line 87: Explain the notation \Xi^{\j} (I assume it is all Xi variables except X_j
  • Line 96: "effects of common factors" are defined with the same notation that "contribution". Is this a different thing? Then use another symbol. If you refer to "contribution" also with "effect", I thing only one term should be used to avoid confusion.
  • Line 97: acronym ECD is not defined. If it is "total relative contribution", put it in the sentence, or whatever each letter stands for.
  • Figure 2: Similarly to Fig. 1, I think subindex X_m should be X_p
  • Figure 2 is not cited in the text (any figure should be cited and described/explained)
  • Line 165: Table 1 are the values of \Delta^T matrix bellow, it can be said.
  • Line 169: I think "are obtained as follows" should be "are the following".
  • Tables 1-4: The "uniqueness" row should be explained at some point, and its interpretation.
  • Line 220: ML is not defined (maximum likelyhood??)

Reviewer 2 Report

Clearly, the mathematical facts and procedures presented in this paper are highly original and useful, with the paper written well. To be pointed out are only the following two points:

[1] Remark 1 appears in both Pages 4 and 5: The remark in Page 5 and the following ones must be renumbered. 

[2] The final paragraph in the abstract (Page 1): "demonstrate the present approach" is unclear, as it does not specify what property of the approach is demonstrated. I think that the phrase "demonstrate the usefulness of the present approach" may be better. In other parts, such unclearness of "demonstrated" properties can be found.        

Reviewer 3 Report

  1. It is nessecary to give comments about normal distribution of the factors ksi_j and epsilon_i. Probably it is connect with data set.
  2. What is g(ksi) in the formulas (4,5). Usually in KL-distance is not member g(ksi). Also, it is nessecary to show the set of integration to ksi.
  3. Theorem 1.  (3) is the "conditonal density function". It is not " factor analysis factor"
  4. The theorem 2 is not right formuletes: what is "A", "decomposition"?
  5. It is nessecary to explane in examples how arise numbers 0.564; 0.193 and other

Round 2

Reviewer 1 Report

Thank you very much for addressing all my concerns and for following my suggestions. The paper has been considerably improved.

I found two typos to be corrected by the editorial team:

  • Line 441: Missing blank space in Table1
  • Equation after line 443: "4.223.CR" should be "4.223, CR"

Congratulations for the work, maybe one day I will assign a student to implement it in R :)